# Single -and Multi-Walled Carbon Nanotubes as Nanocarriers for the Delivery of 7-Hydroxyflavone

**DOI:** 10.3390/pharmaceutics14122806

**Published:** 2022-12-15

**Authors:** Cecilia Espíndola, Alejandro Javier Correa, Manuel López-López, Pilar López-Cornejo, Eva Bernal, José Antonio Lebrón, Francisco José Ostos, Mohammed Rafii-El-Idrissi Benhnia, María Luisa Moyá

**Affiliations:** 1Department of Physical Chemistry, University of Seville, C/Profesor García González 1, 41012 Seville, Spain; caced0167@gmail.com (C.E.); alecoresp@alum.us.es (A.J.C.); evabernal@us.es (E.B.); jlebron@us.es (J.A.L.); fostos@us.es (F.J.O.); 2Department of Chemical Engineering, Physical Chemistry and Material Sciences, Campus de El Carmen, Avda. de las Fuerzas Armadas s/n, 21071 Huelva, Spain; 3Clinical Unit of Infectious Diseases, Microbiology and Parasitology, Institute of Biomedicine of Seville (IBiS), Virgen del Rocío University Hospital, CSIC, University of Seville, 41013 Seville, Spain; rafiim@us.es; 4Department of Medical Biochemistry, Molecular Biology, and Immunology, School of Medicine, University of Seville, 41009 Seville, Spain

**Keywords:** 7-Hydroxyflavone, single-walled carbon nanotubes, multi-walled carbon nanotubes, encapsulation, equilibrium binding constants, drug release

## Abstract

The research on flavonoids has exponentially grown since their first therapeutic evidence, in 1937. They are effective in vitro in a wide range of human diseases, particularly those mediated by free radicals, such as cancer, atherosclerosis, AIDS, or neuronal diseases. However, their applications have been reduced due to their low solubility, poor absorption, and rapid metabolism. Flavonoid encapsulation in nanocarriers significantly improves their oral absorption, protects the drug against degradation, decreases the first-pass hepatic effect, and makes absorption through the lymphatic system easier. In this work, carbon nanotubes were used as nanocarriers of 7-hydroxyflavone, 7-HF. The encapsulation of 7-HF into pristine single- and multi-walled carbon nanotubes, and into -COOH functionalized single-walled carbon nanotubes has been investigated. The equilibrium association constants were estimated. The structural backbone of 7-HF, two benzene rings linked through three carbon atoms that form a pyran heterocyclic ring containing a keto group, seems to play a key role in the 7-HF/CNT interactions, although other types of interactions are also at work. The in vitro release of 7-HF was studied at three pHs, 2.0, 7.4, and 9.2, mimicking the different biological barriers of the human organism.

## 1. Introduction

Flavonoids are polyphenolic phytochemicals, formed from the secondary metabolism of plants. They present a wide range of structures and contribute to the nutritional and organoleptic properties of edible plants, vegetables, fruits, and some beverages, such as coffee, tea, beer, or wine [1,2,3]. Flavonoids can be classified into various groups: flavones, flavanones, flavonoids glycosides, flavonolignans, flavanas, isoflavones, anthocyanidins, aurones, leucoanthocyanidines, neoflavonoids, and chalcones [4]. All flavonoid groups have several therapeutic activities [5], but among them, flavones have been extensively studied due to their anti-inflammatory [4], antimicrobial [6], antioxidant [7], or antitumor activities [8,9]; being used in the treatment of a wide range of human diseases, such as atherosclerosis, diabetes, cancer, Alzheimer’s disease, etc.

7-Hydroxyflavone, 7-HF, has the structure shown in Figure 1. Its basic structural feature is the benzo-c-pyrone (C6–C3–C6) skeleton. This C6–C3–C6 skeleton consists of two aromatic rings (rings A and C) linked through a linear three-carbon chain which formed a closed pyran ring (ring B), where a keto group is present. A hydroxyl group can be found in position 7. Despite 7-HF having a low antioxidant activity measured through methods such as DPPH, 7-hydroxyflavone has been shown to be effective against nicotine-associated oxidative stress [10]. This is so because, although the most frequent mechanism of flavones’ antioxidant activity involves the scavenging of reactive oxygen species (ROS) and peroxynitrite, they can exert an indirect antioxidant activity through the transcriptional induction of genes with antioxidant properties. 7-HF has been found to inhibit lipopolysaccharides-induced inflammation via attenuating the production of NO, prostaglandin E2, PGE2, tumor necrosis factor α, TNF-α, and interleukin 6, IL-6, which are mediators in the inflammatory processes accompanying several diseases such as arthritis [11,12]. 7-HF also exerts a cytotoxic and antiproliferative effect on leukemia H60 cells [13] and inhibits the replication of the enterovirus 71, which causes the hand, foot, and mouth disease, an infection affecting many children in Asia [14]. The neuroprotectant activity of 7-HF, important in relation to aging-associated neurological diseases, has also been investigated [15]. Notwithstanding the therapeutic properties, 7-HF has a low solubility in water, which limits its applications due to poor absorption and rapid metabolism. The encapsulation of this flavone in suitable nanocarriers can substantially increase its solubility, thus improving oral absorption, protecting 7-HF from degradation, and enhancing its therapeutic effects. 

Several types of nanocarriers had been used to encapsulate and, subsequently, release drugs in the therapeutic target [16,17,18,19]. One of those types are carbon nanotubes, CNTs. They are constituted by pure carbon atoms, present in a repetitive hexagonal pattern for cylindrical tubes. They were discovered by Ijima [20] as an allotropic form of carbon by the electric arc discharge of graphitic materials at a high temperature. CNTs can be described as folded forms of graphenes, which are single-layered graphitic sheets. CNTs can be classified into single-walled carbon nanotubes, SWCNTs, double-walled carbon nanotubes, DWCNTs, multi-walled carbon nanotubes, MWCNT, and functionalized carbon nanotubes, f-CNTs [21]. SWCNTs are formed by the folding of one single graphene sheet; in the case of DWCNTs, two sheets of graphene are folded upon each other; and, for MWCNTs, 2 to 10 sheets of graphene are folded into each other or a single sheet is rolled to produce a multi-walled structure (see Figure 2). The functionalization of the CNTs is achieved by a synthetic process with the goal of changing some properties of pristine CNTs, such as water solubility.

CNTs have several characteristics which make them interesting for their use in drug delivery applications: high inner volume, ability for the immobilization of many species, and excellent functionalization ability, accompanied by their biocompatibility [21]. In fact, they have been used in cancer treatment, theranostic applications, and the delivery of therapeutic molecules, such as proteins, peptides, RNA, DNA, siRNA, etc. [22,23,24,25,26]. With this in mind, and with the goal of improving the bioavailability of 7-HF, in this work, the binding of 7-hydroxyflavone to single- and multi-walled carbon nanotubes has been investigated. It is important to point out that, as was mentioned above in the Introduction, 7-HF has important therapeutic properties, which make its encapsulation in adequate nanocarriers an issue worth investigating. The equilibrium association constants were estimated by considering the changes in the flavone emission fluorescence intensity upon increasing the CNTs’ concentration. Pristine MWCNTs and SWCNTs were used, as well as the functionalized SWCNTCOOH. Subsequently, the in vitro release of 7-HF from the CNTs was studied at pHs 2.0, 7.4, and 9.2, mimicking the different biological barriers in a human organism.

## 2. Materials and Methods

### 2.1. Materials

7-Hydroxyflavone (purity > 98%), the organic solvents (purity > 99.9%) used, as well as all the components of the buffer solutions prepared (with the highest purity available), were purchased from Sigma (Darmstadt, Germany). Pristine and functionalized SWCNTs were obtained from NanoLab Inc. (Waltham, MA, USA), both of them with a 1–5 µm length and 1.5 nm diameter. MWCNTs were supplied by Dropsens S.L. (Oviedo, Spain), with a 1.5 µm length and 10 nm diameter.

### 2.2. Preparation of Buffer Solutions

The buffer solution at pH = 9.2 was prepared with NaH_2_PO_4_ and Na_2_HPO_4_, at concentrations of 3.7 mM and 96.3 mM, respectively. The pH was adjusted using a pH-meter Basic 20+ from Crison (Barcelona, Spain).

The neutral buffer solution Tris/HCl, pH = 7.4, was 22.7 mM in tris(hydroxymethyl)aminomethane and 9.8 mM in HCl. The pH was adjusted with the above-mentioned pH-meter.

The acid buffer solution NaCl/HCl, pH = 2.0, was 8.06 mM in NaCl and 1.9 mM in HCl. The pH was adjusted with the above-mentioned pH-meter.

All solutions were prepared with deionized water (from a Millipore Milli-Q system, Darmstadt, Germany), with a conductivity < 10^−6^ S m^−1^.

### 2.3. In vitro Cytotoxicity Assays

The cytotoxicity of the CNTs, at different concentration values, was estimated in vitro using the CyQUANT™ LDH cytotoxicity assay [27]. Several cell lines from a commercial supplier (ATCC^®^, Manassas, VA, USA) were used, Vero E6 (normal monkey kidney epithelial cells), HeLa (human cervical carcinoma epithelial cells), U937 (human leukemia monocytic cells), THP-1 (human leukemia monocytic cells), and Jurkat (human T leukaemia cells). Each cell line was seeded at 10 × 10^4^ cells/well into Nunc flat-bottomed 96-well plates (ThermoFisher Scientific, Waltham, MA, USA), using complete D-10 or R-10 (Dulbecco’s modified Eagle medium (DMEM) or Roswell Park Memorial Institute (RPMI) supplemented with 10% of fetal bovine serum (FBS), and penicillin, streptomycin, and L-glutamine), incubated at 37 °C in 5% CO_2_, and used the following day (75 to 90% confluence). The FBS used in all experiments was heat inactivated (56 °C, 30 min) prior to use to eliminate complement activity. CNT solutions at different concentration values were added to each well and the plates were incubated for 36 h at 37 °C in 5% CO_2_. Controls D-10 or R-10 medium alone were used as the negative control. An amount of 10 µL of 10X Lysis Buffer, and 10 μL of sterile ultrapure water were added to each set of triplicate wells, and used as the Maximum LDH Activity and Spontaneous LDH activity, respectively. Later, the medium from each well was collected by centrifugation of the plate and used to test the cytotoxicity of the CNT solutions using the CyQUANT™ LDH Cytotoxicity Assay Kit according to the manufacturer’s instructions (Invitrogen™ from Thermo Fisher Scientific, Waltham, MA, USA). The cytotoxicity was measured by fluorescence in a CLARIOstar^®^ (BMG LABTECH, Allmendgrün, Ortenberg, Germany). Each CNT concentration was measured in triplicate and the tests were repeated thrice independently. The cell viability was calculated by using Equation (1): 
(1)
$$\%\;\mathrm{Cell}\;\mathrm{viability}=100-\left(\left[\frac{\mathrm{Compound}-\mathrm{treated}\;\mathrm{LDH}\;\mathrm{activity}-\mathrm{Spontaneous}\;\mathrm{LDH}\;\mathrm{activity}}{\mathrm{Maximum}\;\mathrm{LDH}\;\mathrm{activity}\;-\;\mathrm{Spontaneous}\;\mathrm{LDH}\;\mathrm{activity}}\right]\times 100\right)$$


Cell viability values were also checked by trypan blue method [28] and no significant differences were observed.

### 2.4. Fluorescence Measurements

Fluorescence emission spectra of 7-HF were registered in a Hitachi 2500 spectrofluorimeter, connected to a Lauda water flow thermostat to keep the temperature constant at 298.0 ± 0.1 K. A quartz cell of 10 mm path length was used.

The excitation wavelength was λ = 350 nm and the spectra were recorded from λ = 400 to λ = 650 nm. The maximum fluorescence emission intensity was measured at λ = 526.nm. The excitation and emission slits were 2.5 and 5 nm, respectively.

#### 2.4.1. Fluorescence Emission Intensity Calibration Curves

A concentrated 7-HF solution, 5.04·10^−4^ M, in ethanol was prepared. This solution was kept at 4 °C in the dark in order to avoid the flavone degradation. Three calibration curves were obtained, each one for the three different pH conditions. 7-HF solutions of known concentrations, within the range 2.52·10^−7^–2.02·10^−5^ M, were prepared as follows. An adequate aliquot of the ethanolic concentrated 7-HF solution was added to a 10 mL flask. The organic solvent was evaporated with an air flow and, subsequently, the buffer was added up to the calibration mark. The flask was then introduced in a sonicator for 20 min, in darkness. In this way, the flavone is totally dissolved in the buffer solution. Once the fluorescence emission intensity of each solution is measured at λ = 526 nm, the calibration curve is obtained. Temperature was kept at 298.0 ± 0.1 K.

### 2.5. Encapsulation of 7-HF in the CNTs

The encapsulation of 7-HF in the different CNTs investigated was carried out at pH = 7.4 (buffer Tris/HCl), keeping the flavone concentration constant at 1.50·10^−5^ M and varying the CNT concentration within the range 0–0.35 g/L for the SWCNTs, 0–0.22 g/L for the MWCNTs, and 0–0.10 g/L for the SWCNT-COOH. First, an adequate aliquot of the ethanolic flavone solution was added to an Eppendorf tube of 2 mL. After evaporating the organic solvent with an air flow, 2 mL of a CNT solution of a known concentration, in a Tris/HCl buffer, was added to the Eppendorf tube. Afterwards, the tube was sonicated for 20 min in darkness. Finally, the tube was centrifuged at 13,500 rpm for 30 min. Once the supernatant was separated, the fluorescence emission intensity was measured, at 298.0 ± 0.1 K, under the working conditions indicated in Section 2.4. In this way, the amount of 7-HF not bound to the CNTs can be known by using the calibration curve at pH = 7.4.

### 2.6. Encapsulation Efficiency

The encapsulation efficiency was calculated by using the following equation:
(2)
$$\%7-HF_{encapsulated}=\left(1-\frac{m_{7-HF}^{supernatant}}{m_{7-HF}^{total}}\right)\cdot 100$$

where 
$m_{7-HF}^{supernatant}$
 is the amount of the flavone left in the supernatant when the 7-HF bound to the CNTs was removed, and 
$m_{7-HF}^{total}$
 is the total amount of flavone initially added in the Eppendorf tube. 
$m_{7-HF}^{supernatant}$
 was estimated by measuring the fluorescence emission intensity at λ = 526 nm and 298.0 ± 0.1 K, as was mentioned above.

### 2.7. Zeta Potential Measurements

The zeta potential, ξ, was determined by using a Zetasizer Nano ZS Malvern Instrument Ltd. (UK). The samples were diluted with filtered water to an adequate concentration. A scattering angle of 90° was used in the size distribution analysis. All measurements were carried out at 298.0 K. At least six measurements were obtained for each sample and the average value (standard deviation) was considered.

### 2.8. In Vitro 7-HF Release

With the goal of studying the 7-HF release from the CNTs, a method described elsewhere was used [29]. An adequate aliquot of the 7-HF ethanolic solution was added to a 2 mL flask. The organic solvent was evaporated with an air flow. Afterwards, a buffer solution was added up to the calibration mark. Then, the flask, in darkness, was sonicated for 20 min to make sure all the flavone was dissolved in the buffer solution. The 7-HF concentration in the buffer solution was 2.52·10^−4^ M. This flavone solution will be called solution A.

In a 2 mL Eppendorf tube, 2 mg of CNTs were weighted and then, 2 mL of solution A were added to the tube, which was sonicated for 30 min in darkness and, subsequently, it was centrifuged at 13,500 rpm for 30 min. The supernatant was separated and the CNTs with the bound 7-HF were dried under vacuum by using an Eppendorf Concentrator Plus from Eppendorf Ibérica (Madrid, Spain) for 45 min. The 7-HF concentration in the supernatant was estimated measuring the fluorescence emission intensity of the solution at λ = 526 nm and using the corresponding calibration curve, depending on the pH of the medium. In this way, the amount of 7-HF not bound to the CNTs can be estimated and, therefore, that of the CNTs’ associated 7-HF (7-HF/CNT). Finally, 1.8 mL of the buffer solution was added to 10 mg of the dried 7-HF/CNT complexes within the Eppendorf tube, which was maintained under continuous magnetic stirring (200 rpm) in darkness. At determined time intervals, approximately 24 h, the tube was centrifuged at 13,500 rpm for 30 min. Subsequently, 1 mL of supernatant was removed, and the same volume of the corresponding buffer replaced in the Eppendorf tube. This is a way of mimicking the in vivo removal into the systemic circulation. The Eppendorf tube was put under continuous magnetic stirring, always in darkness, to avoid 7-HF photodegradation. The emission fluorescence intensity of the supernatant sample was measured at 526 nm in order to calculate the flavone concentration using the corresponding calibration curve.

Each experiment was conducted by triplicate. In addition, the release was studied at pHs 2.0, 7.4, and 9.2, simulating the different biological barriers in our organism.

### 2.9. Statistical Analysis

Values are expressed as the mean ± standard errors of separate experiments. Statistical analysis was performed with Student’s *t*-test and One-way analysis of variance (ANOVA). When *p* < 0.05 (95% confidence), the differences were considered as significant.

## 3. Results and Discussion

### 3.1. Fluorescence Emission Intensity Calibration Curves for 7-HF at Different pHs

The fluorescence emission intensity calibration curves at different pHs were obtained as detailed in Section 2.4.1. They are shown in Figure 3. In all cases, straight lines with good correlation coefficients were observed.

### 3.2. Cytotoxicity of CNTs

It is important to have information about the toxic character of any nanosystem being used as a drug carrier. Bearing this in mind, the cytotoxicity measurements of the SWCNT, MWCNT, and MWCNT were investigated using the cell lines Vero E6 (normal monkey kidney epithelial cells), HeLa (human cervical carcinoma epithelial cells), U937 (human leukemia monocytic cells), THP-1 (human leukemia monocytic cells), and Jurkat (human T leukaemia cells). The results are shown in Figure 4.

Figure 4 shows that the CNTs are non-toxic for the normal cell line Vero E6, even for concentrations well above those used in this work. The SWCNTs begin to be toxic for the U937 and Jurkat lines for [SWCNT] > 0.156 g L^−1^, although at higher concentrations the toxicity is highest for the HeLa cancer line. The THP-1 cancer cell line is not much affected by the presence of these carbon nanotubes. The MWCNTs are non-toxic for the normal cell lines within the concentration range studied, as in the case of the SWCNTs. The former are particularly toxic for the U937 cell line for [MWCNT] > 0.156 g L^−1^. For subsequent increases in [MWCNT], they are also toxic for the Jurkat cell lines and, only at high MWCNT concentrations, the CNTs are toxic for the HeLa cell line. As in the case of the SWCNT, the THP-1 is practically not affected for the MWCNTs. With regard to the SWCNTCOOH, these carbon nanotubes are non-toxic, within a wide concentration range, for all the cell lines investigated.

### 3.3. Encapsulation of 7-HF in CNTs

Following the methodology described in Section 2.4 and using Equation (2), it is possible to calculate the amount of flavone encapsulated for each CNT concentration at pH = 7.4. Figure 5 shows the dependence of I/I_o_ on the CNT concentration, where I and I_o_ are the emission fluorescence intensities at *λ* = 526 nm of the 7-HF solutions in the presence and in the absence of the CNTs, respectively. Since the encapsulation percentage of 7-HF bound to the CNTs is proportional to I/I_o_, the data in Figure 4 indicate that the amount of 7-HF associated depends on [CNTs], following a sigmoidal function for the three types of CNTs investigated. The sigmoidal dependence of I/I_o_ on the CNT concentration indicates that the association between 7-HF and the CNTs is cooperative. This means that the binding of one 7-HF molecule favors the association of a new flavone molecule to the CNTs and so on [30]. Table 1 summarizes the CNT concentrations for which a complete encapsulation of 7-HF is reached. At this point, it is worth noting that the interaction is expected to occur between the aromatic rings of the flavone and the cloud of π electrons at the CNT surfaces. However, since these interactions would be operative in all the CNTs studied, the differences observed point out that other interactions should be at work. This issue will be considered below.

With the goal of quantifying the 7-HF/CNT interactions, the equilibrium binding constant of 7-HF to the three types of CNTs, K, was calculated. The equilibrium can be written as:
(3)
7−HF+CNTs ⇌7−HF/CNT


Therefore: 
(4)
$$K=\frac{\left[7-\mathrm{HF}/\mathrm{CNT}\right]}{\left[7-\mathrm{HF}\right]\left[\mathrm{CNT}\right]}\;$$


The dependence of I/I_o_, on [CNTs] can be expressed, considering the Pseudophase Model, as follows [31]: 
(5)
$$I/I_{o}=\frac{{\left(I\right)}_{f}+{\left(I\right)}_{b}K\left[CNT\right]}{1+K\left[CNT\right]}$$

where 
${\left(I\right)}_{f}$
 and 
${\left(I\right)}_{b}$
 are the fluorescence emission intensity of 7-HF free in the solution and bound to the CNTs, respectively. In a cooperative interaction, K depends on [CNT], increasing upon augmenting the CNT concentration, until reaching a maximum value, the limiting value K_max_. Taking this into account, K can be written as [31]:
(6)
$$K=\frac{K_{max}\cdot \;e^{t}}{1+e^{t}}$$

where t is given by Equation (6):
(7)
$$t=\frac{\left[\mathrm{CNT}\right]-h}{j}$$


In this equation, K_max_ is the maximum value reached by K, h is the CNT concentration, for which K = K_max_, and j is an adjustable parameter.

The dependence of I/I_o_ on the CNT concentration shown in Figure 4 was fitted by using Equations (5)–(7). The result of the fittings is represented by the solid lines in this Figure. One can see that the agreement between the experimental and the theoretical data is good. The limit K_max_ values obtained from the fittings are listed in Table 2.

The estimated K_max_ values are similar, within experimental errors, for the three CNTs investigated. The fact that K is different, at the same CNT concentration, for the different carbon nanotubes investigated, explained the observed trend [SWNCT]_100 7%-HFencapsulated_ > [MWNCT]_100% 7-HFencapsulated_ > [SWNCTCOOH]_100 7%-HFencapsulated_ (Figure 5 and Table 1). A plausible explanation to rationalize the differences observed between pristine single- and multi-CNTs, could be related to the length and diameter of both types of nanotubes. The SWCNTs have a length of 1–5 µm and a diameter of 1.5 nm, whereas the MWCNTs have a length of 1.5 µm and a diameter of 10 nm. In addition, the latter have multiple inner layers. Taking this into account, a higher number of 7-HF molecules would be expected to be bound to one MWCNT molecule than to an SWCNT one. As a consequence, [SWNCT]_100 7%-HFencapsulated_ would be higher than [MWNCT]_100% 7-HFencapsulated_, in agreement with the observations. In relation to the high affinity shown by the flavone for the SWCNTCOOH carbon nanotubes, at pH 7.4, most of the COOH groups would be ionized [32]. Therefore, these CNTs would be negatively charged. On the other hand, the 7-HF molecules are expected to be mainly non-ionic in water at pH 7.4 [33], although some anionic form is present. Maybe, the increase in the charge density at the CNT surface will favor the interactions with the aromatic rings of 7-HF, thus explaining the low SWCNT-COOH concentration necessary to completely bind the 7-HF. Additionally, the functionalization of the nanotube with carboxylic groups favors the formation of hydrogen bonds with the –OH groups of 7-HF, increasing the affinity of the flavone for the nanotubes.

The stability of the 7-HF/CNT complexes was checked using zeta potential measurements. The results were ξ = −10 ± 3 for the 7-HF/SWCNT complexes, ξ = −15 ± 5 for the 7-HF/MWCNT complexes, and ξ = −31 ± 7 for the 7-HF/SWCNTCOOH complexes. These values did not change, within experimental errors, for various days.

### 3.4. Study of the In Vitro 7-HF Release

The in vitro release of 7-HF from the 7-HF/CNT complexes was studied following the procedure described in Section 2.3. Since most flavonoids, and 7-HF in particular, are orally administered to patients, the 7-HF release was investigated at different pHs, mimicking the conditions of the stomach, large intestine, and small intestine; that is, pHs 2.0, 7.4, and 9.2, respectively. In addition, pH 7.4 is also the physiological pH, which is that which the 7-HF/CNT complexes will encounter after its absorption in the small intestine.

From the fluorescence emission intensity values, the amount of 7-HF released from the 7-HF/CNT complexes can be calculated as:
(8)
$$\%7-HF_{released}=\frac{{\left[7-HF\right]}_{\;accumulated}^{t}}{[7-HF{]}_{total}}\cdot 100$$

where 
${\left[7-HF\right]}_{\;accumulated}^{t}$
 is the flavone concentration released after a time t and 
$[7-HF{]}_{total}$
 is the total flavone concentration initially present in the Eppendorf tube (containing 10 mg of the 7-HF/CNT complex and 1.8 mL of buffer). Figure 6 shows the release profiles at the three pHs for the 7-HF/SWCNT complexes; that is, the dependence of the percentage of the flavone, initially encapsulated, which has been released, %7-HF_released_, against time.

In all the buffer solutions used, the release of 7-HF was followed for at least 400 h. After this time, the flavone release in neutral and basic media is really slow. However, after 400 h, in acid medium, %7-HF_released_ continues increasing with time. Taking the acid-based properties of 7-HF into account [33], at pH 2.0, all the flavone molecules are in their non-ionic form. At pH 7.4, most of the flavone is in its non-ionic form, but the anionic form is also present. At pH 9.2, the 7-HF anionic form concentration is twice or three times higher than that of the non-ionic form. At first, different interactions between the non-ionic and anionic forms of the flavone with the CNTs would be expected, thus resulting in different release rates for the two flavone forms [33]. Therefore, a one-phase release profile was expected in acid medium, whereas a two-phase release profile should be observed in neutral and basic media. Figure 6b,c show a two-phase release profile, in agreement with the expectations. Not enough data were available in acid medium. Even so, in order to further investigate this issue, a kinetic analysis of the release profiles has been performed. Data in Figure 5a, acid medium, were fitted using Equation (9) and data in Figure 5b,c, neutral and basic media, were fitted by using Equation (10).

(9)
$$\%7-HF_{released}=a\cdot \left(1-exp\left(-k_{n}\cdot t\right)\right)$$


(10)
$$\%7-HF_{released}=a\cdot \left(1-exp\left(-k_{n}\cdot t\right)\right)+b\cdot \left(1-exp\left(-k_{a}\cdot t\right)\right)$$


Here, k_n_ and k_a_ are the first order rate constants corresponding to the release of the non-ionic 7-HF and anionic 7-HF forms, respectively. *a* and *b* are two adjustable parameters related to the released 7-HF in its non-ionic and anionic forms, respectively. The first order rate constants obtained from the fittings are listed in Table 3. The solid lines in Figure 5 are the result of the fittings. The agreement between the experimental and the theoretical values was good. The red dots in Figure 6a were calculated by using the adjustable parameters estimated from the fitting by using Equation (9).

Table 3 shows that the estimated k_n_ and k_a_ values are the same in the different buffer solutions, within experimental errors, as was expected. One can see that k_n_ < k_a_; that is, the release rate of the anionic form from the 7-HF/SWCNT complexes is faster than that of the non-ionic form. This could explain the results shown in Figure 6. In acid medium, only non-ionic 7-HF molecules are present. After their association with the CNTs, the subsequent release follows a one-phase profile. This release is really slow. In fact, the estimated time to reach the maximum value of %7-HF_released_ would be longer than 2000 h. In neutral and basic media, the non-ionic and anionic flavone forms are present in the solution. Once they are bound to the CNTs, the release presents a two-phase profile. Taking into account the percentages of flavone anionic form present in the solution at pH 7.4 and 9.2, the amount of flavone released, within the same period of time, would be expected to be larger in basic medium than in neutral medium since k_a_ > k_n_.

In relation to the flavone release from the 7-HF/MWCNT complexes, after following the release for more than 400 h, negligible amounts of %7-HF_released_ were found, at the three pHs investigated. The same result was found for the SWCNT-COOH carbon nanotubes. This would point out that the release rate constants, for both the non-ionic and the anionic flavone forms, k_n_ and k_a_, would be slower in pristine MWCNTs and functionalized single-walled carbon nanotubes, SWCNT-COOH, in spite of the K_max_ being similar for the three CNTs investigated.

## 4. Conclusions

In the present work, the encapsulation of 7-hydroxyflavone, 7-HF, within pristine single- and multi-walled carbon nanotubes, SWCNTs and MWCNTs, respectively, and within functionalized single-walled carbon nanotubes, SWCNT-COOH, has been investigated. The results show that the binding was cooperative and that 100% of the flavone was encapsulated by the CNTs. In addition, the K_max_ values estimated for the binding of 7-HF to the three CNTs studied were similar, within experimental errors. However, the CNT concentration necessary to completely bind the flavone is different for the three carbon nanotubes, following the trend [SWNCT]_100 7%-HFencapsulated_ > [MWNCT]_100% 7-HFencapsulated_ > [SWNCTCOOH]_100 7%-HFencapsulated_.

A plausible explanation for rationalizing that [SWNCT]_100 7%-HFencapsulated_ > [MWNCT]_100% 7-HFencapsulated_ could be that the MWCNTs present a larger binding surface than the SWCNTs. Therefore, a larger number of 7-HF molecules would be associated to one pristine neutral multi-walled CNT molecule than to an SWCNT one. The experimental observation that the lower CNT concentration necessary to bind 100% of the flavone molecules corresponds to the SWCNT-COOH could be related to a more favorable 7-HF/CNT association when the negative charge density at the CNT surface increases. One has to consider that, at pH 7.4, most of the functionalized CNTs are negatively charged, but the majority of the 7-HF molecules are non-ionic. In addition, the functionalization of the nanotube with carboxylic groups favors the formation of hydrogen bonds with the –OH groups of 7-HF, increasing the affinity of the flavone for the nanotubes.

All the CNTs used as nanocarriers in this work are non-toxic for normal cell lines within a wide concentration range.

The in vitro release of 7-HF from the 7-HF/CNT complexes was followed for a long period of time at three different pHs, 2.0, 7.4, and 9.2. These pH values were chosen for mimicking the biological barriers those complexes will encounter when they are orally administered, which is the usual way of flavonoid administration.

The release of the flavone from the 7-HF/SWCNT complexes shows a biphasic profile in neutral and basic media, whereas in acid medium, a monophasic one was found. This was explained by considering that, in acid medium, only the non-ionic 7-HF form was present in the solution, while the non-ionic and anionic flavone forms are present in neutral and basic media. After a kinetic analysis, the first order rate constants corresponding to the non-ionic and anionic flavone forms release, k_n_ and k_a_, were estimated by fitting the experimental data to the appropriate equations. It was found that k_n_ < k_a_. This result explains that the maximum %7-HF_released_ was found in basic pH, since at this pH the highest amount of anionic flavone form is present.

In the case of the MWCNTs and SWCNT-COOHs, a negligible amount of flavone was released after more than 400 h. This could be due to the first order rate constants k_n_ and k_a_ being lower in these CNTs than in the SWCNTs.

From the results mentioned above, one can conclude that SWCNTs can be used as nanocarriers for the oral administration of 7-HF.

## Figures and Tables

**Figure 1 pharmaceutics-14-02806-f001:**
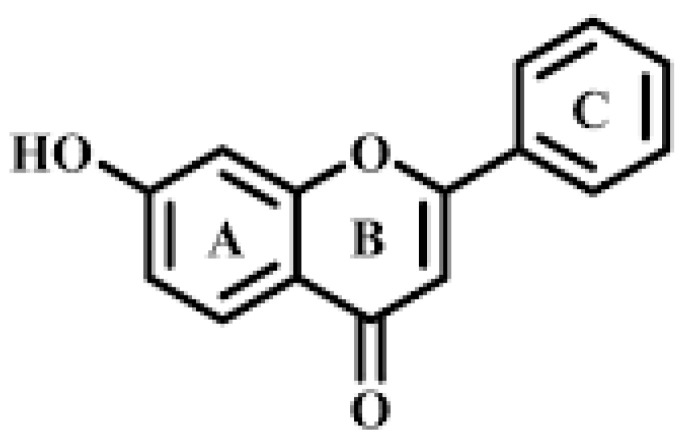
Structure of 7-hydroxyflavone, 7-HF.

**Figure 2 pharmaceutics-14-02806-f002:**
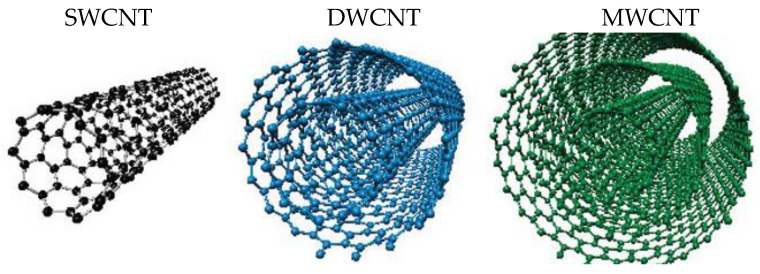
Structure of carbon nanotubes, CNTs.

**Figure 3 pharmaceutics-14-02806-f003:**
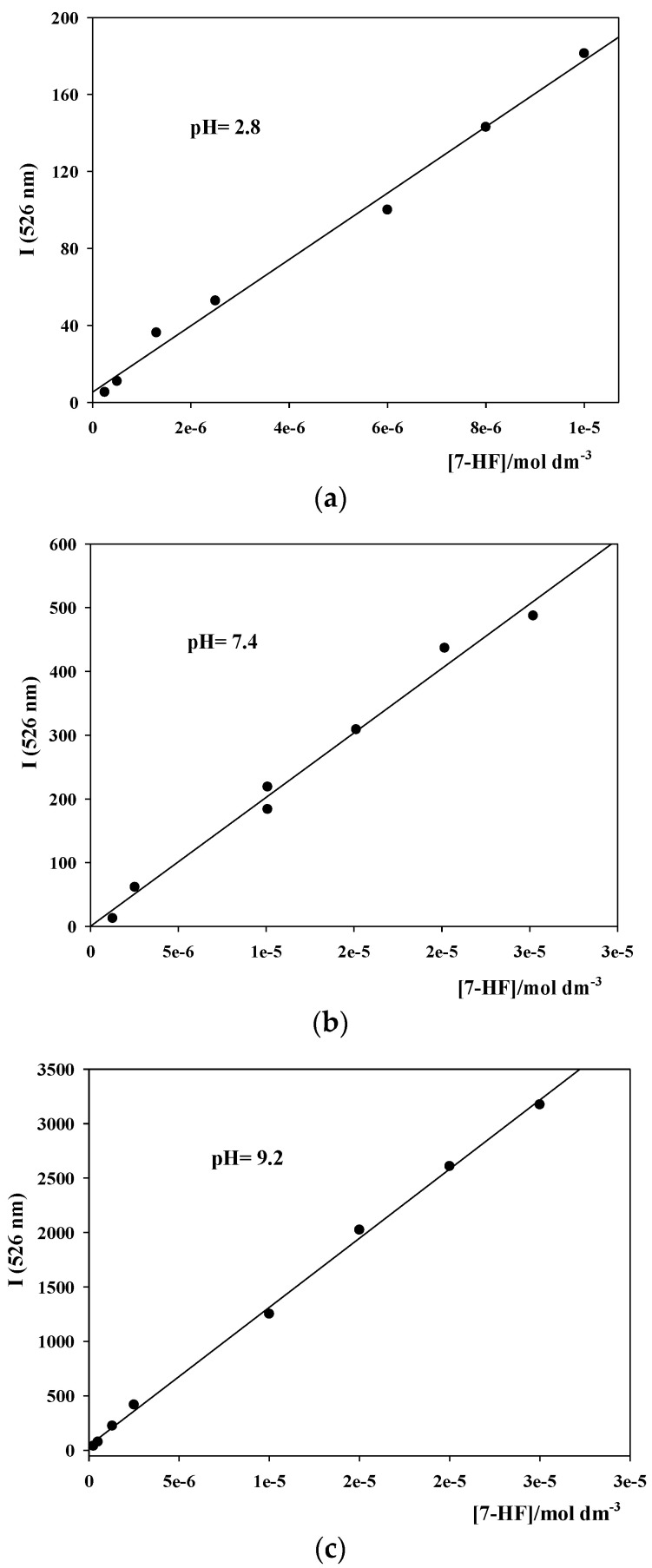
Fluorescence emission intensity calibration straight lines for 7-HF at different pHs. (**a**) pH = 2.0; (**b**) pH = 7.4; (**c**) pH = 9.2. T = 298.0 ± 0.1 K.

**Figure 4 pharmaceutics-14-02806-f004:**
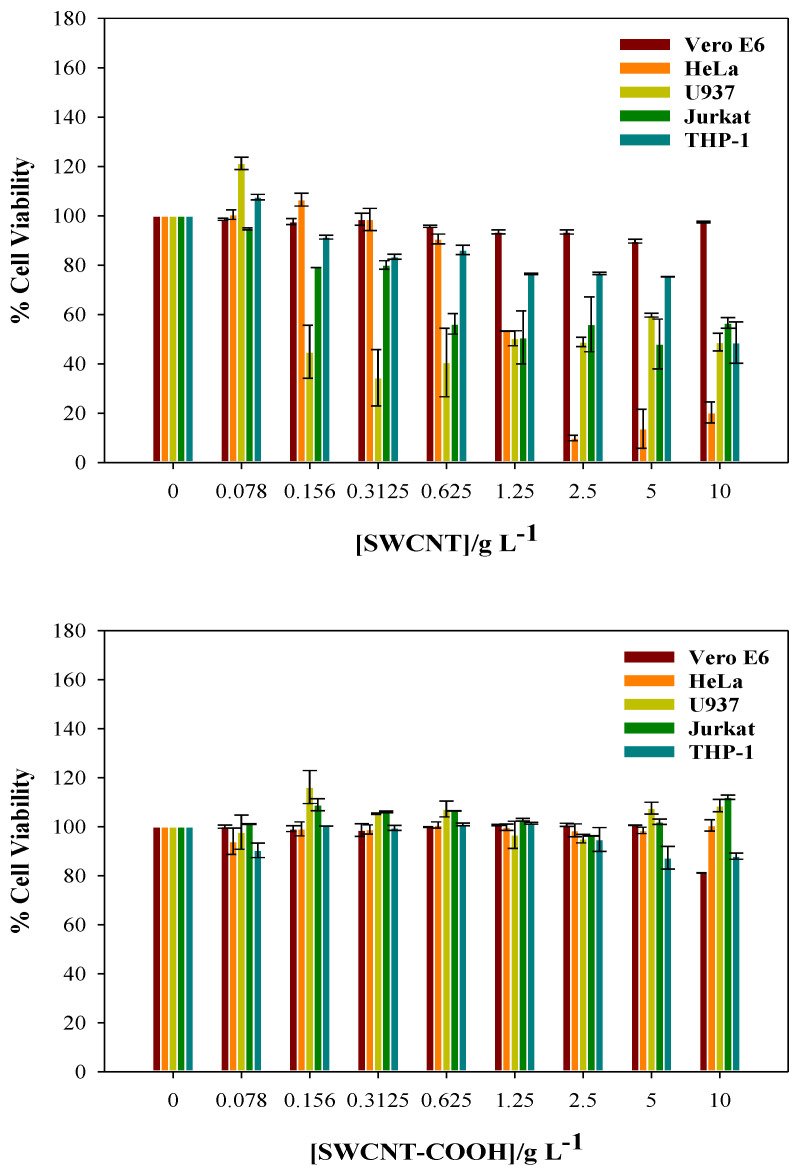
Cell viability of different carbon nanotubes in normal and cancer cell lines.

**Figure 5 pharmaceutics-14-02806-f005:**
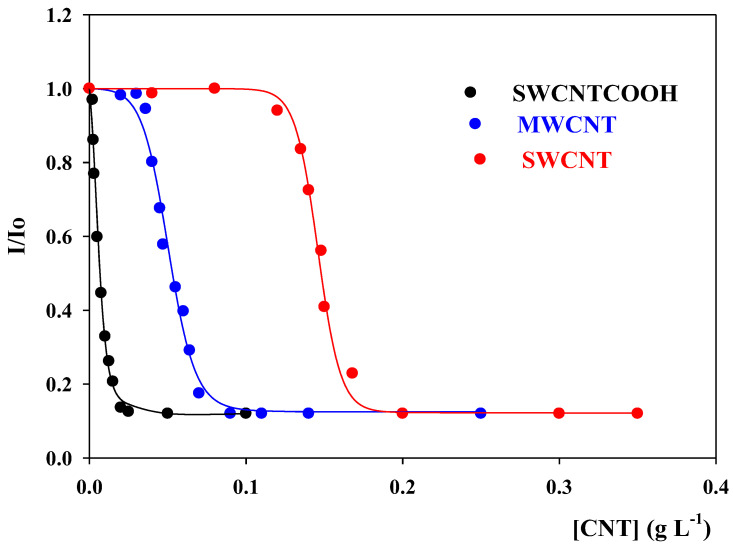
Dependence of I/I_o_ (526 nm) on CNT concentration at pH = 7.4. Solid lines are the result of fitting the experimental data to Equations (4) and (5). T = 298.0 ± 0.1 K.

**Figure 6 pharmaceutics-14-02806-f006:**
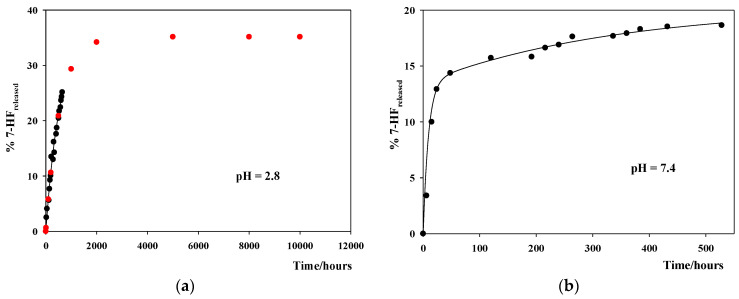
In vitro release profiles of 7-HF from 7-HF/SWCNT complexes at different pHs. (**a**) pH = 2.0; (**b**) pH = 7.4; (**c**) pH = 9.2. Solid lines correspond to the fitting of the experimental data using Equation (8), for pH = 2.0, and Equations (8) and (9), for pHs 7.4 and 9.2. The red dots are the predicted values for the complete release profile at pH = 2.0. These values were obtained by using Equation (8).

**Table 1 pharmaceutics-14-02806-t001:** CNT concentration necessary to completely encapsulate the 7-HF at pH = 7.4.

	100% 7-HF_encapsulated_		
pH = 7.4	[SWNCTs]/g·L^−1^	[SWCNT-COOH]/g·L^−1^	[MWNCTs]/g·L^−1^
[7-HF] = 1.51·10^−5^ M	0.20	0.025	0.090

**Table 2 pharmaceutics-14-02806-t002:** Values of K_max_ obtained by fitting the I/I_p_ data shown in Figure 4 using Equations (5)–(7). T = 298.0 ± 0.1 K.

CNTs	K_max_/g^−1^ L
SWCNT	(1.5 ± 0.3)·10^3^
MWCNT	(1.2 ± 0.2)·10^3^
SWCNT-COOH	(1.5 ± 0.3)·10^3^

**Table 3 pharmaceutics-14-02806-t003:** First order rate constants obtained from the fittings of the dependence of %7-HF_released_ on time, using Equation (9) for the data in acid medium, and Equation (10) for the data in neutral and basic media. The data correspond to the release of 7-HF from the SWCNT/7-HF complex. T = 310.0 ± 0.1 K.

**7-HF/SWCNT**		
**pH**	**10^3^ k_n_/h^−1^**	
2.0	1.7 ± 0.4	
**pH**	**10^3^ k_n_/h^−1^**	**10^3^ k_a_/h^−1^**
7.4	1.8 ± 0.5	68 ± 10
9.2	1.9 ± 0.7	50 ± 14

## Data Availability

Data is contained within de article.

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
