# Peer review of "Single -and Multi-Walled Carbon Nanotubes as Nanocarriers for the Delivery of 7-Hydroxyflavone"

_pharmaceutics, 2022, doi:10.3390/pharmaceutics14122806_

Round 1
Reviewer 1 Report
1. Kindly add references in the method of estimation in vitro release of 7-HF
2. kindly cite following article Liang Y, Xie M, Li J, Liu L, Cao Y. Influence of 3-Hydroxyflavone on Colloidal Stability and Internationalization of Ag Nanomaterials Into THP-1 Macrophages. Dose-Response. 2019;17(3). doi:10.1177/1559325819865713
Reviewer 2 Report
The manuscript entitled, ‘Single- and multi-walled carbon nanotubes as nanocarriers for the delivery of 7-hydroxyflavone’ reported delivery of therapeutic load via CNTs modified delivery vehicles. There are some loopholes which should be accounted before publication;
1. Why 7-hydroxyflavone is significant for delivery is not clear. That should be emphasized.
2. Is this attachment is versatile for all types of drugs or for some specific drug molecules?
3. The system looks like physical capturing. In that case up to what extent the attachment was robust enough to be stable in different body simulant conditions?
4. Did the author check the long term stability of the system or surface charges of the whole system?
5. The release PL spectra are not visible. That would be better to understand rather than their calibration plots.
6. CNTs are carcinogenic especially for in vivo applications. What measures did the author take here? What about the toxicity?
7. Do the drug-attached CNTs show higher encapsulation? Did the author check on that?
8. There are several articles which have immense significance; some recommendations are here: DOI: 10.1007/978-981-16-8146-2_6; https://doi.org/10.3390/pharmaceutics13040587; https://doi.org/10.3390/polym13234259; https://doi.org/10.1016/j.jddst.2021.102426.
Author Response
Please seed the attachment

Round 2
Reviewer 2 Report
It can be accepted in its present state.